# Long-Term Impact of Domestic Violence on Individuals—An Empirical Study Based on Education, Health and Life Satisfaction

**DOI:** 10.3390/bs13020137

**Published:** 2023-02-07

**Authors:** Liu Bo, Peng Yating

**Affiliations:** 1School of Economics, Hunan Agricultural University, Changsha 410128, China; 2School of Pharmacy, Changsha Health Vocational College, Changsha 410600, China

**Keywords:** domestic violence, emotional abuse, educational achievements, physical and mental health

## Abstract

This paper takes the China Health and Retirement Longitudinal Survey (CHARLS) as a sample to assess the long-term impacts of domestic violence experienced in childhood on individuals. First, from the four dimensions of injury from violence, negligent care, emotional abuse and witness to domestic violence, an indicator system for quantifying domestic violence is constructed. Second, the simultaneous equation of self-evaluation health and life satisfaction is estimated by the seemingly unrelated regression model. Starting with education, health and life satisfaction, the long-term impact of domestic violence experiences on individuals is quantitatively assessed, providing empirical evidence for preventing and curing domestic violence and healing trauma. The empirical research shows the following: (1) An experience of domestic violence significantly reduces educational achievements. Compared with the three dimensions of injury from violence, negligent care and witnessing domestic violence, emotional abuse has the greatest negative impact on educational achievements. (2) Domestic violence significantly reduces the self-assessed health level and life satisfaction and increases the subjective mental health risk. Based on the complexity and concealment of domestic violence, combined with empirical research conclusions, this paper proposes countermeasures to prevent and control domestic violence.

## 1. The Raising of Questions

“Almost the love of children, love and fear arrest, such as the beginning of vegetation germination, ease of the bar, the destruction of the impotence” (Wang Yangming’s “General Idea of Discipline”). Domestic violence is one of the most negative experiences that can impact the temperament of teenagers, and the trauma it brings may accompany them for life. For a long time, news about domestic violence has frequently been reported. How to prevent and control domestic violence is a key issue in governance and public opinion. On 1 March 2016, the “Anti Domestic Violence Law of the People’s Republic of China” (hereinafter referred to as the “Anti Domestic Violence Law”) was officially implemented, allowing the state to directly intervene in domestic violence through special laws. On 23 October 2021, the “Family Education Promotion Law of the People’s Republic of China” was officially promulgated, which further clarified that “parents or other guardians of minors shall not discriminate against minors on the basis of sex, physical condition, intelligence, etc., and shall not commit domestic violence”. With the joint efforts of the government, society and the media, remarkable results in the prevention and treatment of domestic violence have been achieved, but how to heal the trauma caused by domestic violence still needs to be explored. Adler, a famous psychologist, believes that “an unhappy childhood needs a lifetime to be cured” [1]. Trauma events can have a broad and lasting impact on individuals, and clarifying the long-term impact of domestic violence on individuals is a prerequisite for healing the trauma [2,3].

For minors, domestic violence refers to the information perceived by minors that is related to violence in the family and can be divided into direct exposure and indirect exposure according to the form of violence exposure. The former refers to direct physical attacks and abuse suffered by minors at home, while the latter refers to the violence or aggressive behavior of family members that is seen or heard by minors [4,5,6].

In the existing laws and conventions, the definition of the scope of domestic violence is not completely unified. Article 19 of the United Nations Convention on the Rights of the Child stipulates: “When a child is under the care of parents, legal guardians or any other person responsible for the care of the child, he or she shall be protected from any form of physical or mental abuse, injury or abuse, neglect or improper care, abuse or exploitation”. Article 2 of Japan’s “Child Abuse Prevention Law” stipulates that “corresponding to the obvious abuse or rejection of children, illegal attacks such as physical violence against the spouse of a family with children living together endanger their lives or bodies and other words and deeds that have significant psychological harm to children.” There are plans for domestic violence, physical abuse, neglect, emotional abuse and other behaviors to be included in the category of domestic violence [7,8]. Article 2 of the “Anti Domestic Violence Law” of the People’s Republic of China stipulates that “domestic violence referred to in this Law refers to physical and mental violations committed among family members by means of beating, binding, maiming, restricting personal freedom, as well as constant abuse and intimidation.” Therefore, some scholars believe that negligent care, emotional abuse and the witnessing domestic violence should be defined as domestic violence against minors based on the distinctiveness of minors [9,10,11,12].

No matter what the form of domestic violence is, it will cause physical and psychological trauma to minors. Empirical evidence shows that domestic violence seriously harms children’s growth, and its cumulative effects may last until adulthood [13,14,15,16]. The harm caused by domestic violence is different for children of different ages, and early and long-term contact may cause more serious problems [17]. For preschool children and school-age children whose mothers have experienced domestic violence during pregnancy, 44% of them have at least one trauma symptom and separation anxiety [18]. It is often witnessed that domestic violence affects the brain development of children [19]. Lundy and Grossman (2005) [20] conducted a sample survey of 4636 children who had experienced domestic violence. One-fifth of them found it difficult to abide by school rules, and one-third of them were highly aggressive. This conclusion was also confirmed in another survey [21]. The harm caused by witnessing domestic violence cannot be ignored. Compared with children who have not witnessed domestic violence, preschool children who have witnessed domestic violence are more likely to have post-traumatic stress symptoms and find it more difficult to cultivate empathy and inferiority [22,23]. Similar to children, adolescents exposed to domestic violence are more likely to have various psychological and physical problems, experience sleep or eating disorders, engage in drug and alcohol abuse and are more likely to become perpetrators and victims of domestic violence in adulthood [24,25].

The existing literature has examined the definition and category of domestic violence from the perspective of the law, analyzed the adverse impact of domestic violence on personal growth from the perspective of psychology and proposed governance strategies regarding domestic violence from the perspective of social governance, but quantitative research is lacking. This paper uses the CHARLS (2011, 2013, 2015, 2018) and the “life course” survey as sample data to quantitatively assess the long-term impacts of the domestic violence experience on individuals from the perspectives of education, health and life satisfaction. The original intention of this paper is to provide empirical evidence to prevent domestic violence and heal trauma.

This paper consists of four parts as follows: first, based on the life course survey data of the CHARLS, we select the dimensions and indicators to quantify domestic violence and build an empirical model; second, we estimate the empirical model with sample data and adjust the empirical model to test the robustness of the empirical conclusion; finally, the research conclusions are summarized, and the corresponding countermeasures and suggestions are proposed.

## 2. Research Design

### 2.1. Measurement of Domestic Violence

This paper uses the data of China Health and Retirement Longitudinal Survey (CHARLS) from 2011 to 2018 (as shown in https://g2aging.org accessed on 13 December 2022). CHARLS survey was carried out in 2011, 2013, 2015 and 2018. The sample covered 150 counties, 450 communities (villages) and 12,400 households in 28 provinces (autonomous regions, municipalities directly under the Central Government), with 19,000 respondents. The survey conducted four levels of sampling when selecting samples. PPS probability sampling proportional to the population size was adopted in the county (district) village (resident) sampling and then randomly selected sample households from each sample village/neighborhood committee through field mapping. A family member over 45 years of age was randomly selected from each sample household as the main interviewee to interview him (her) and his/her spouse; therefore, the accuracy, unbiased and representativeness of samples are guaranteed. CHARLS provides a wealth of personal, family and community information, including demographic variables and health information at the individual level, wealth, assets, occupation and income variables at the family level and financial and economic development variables at the community level [26,27]. In particular, CHARLS conducted a detailed survey on whether the interviewees suffered from domestic violence and bullying in their childhood and collected information on 12 bad childhood experiences and 14 chronic diseases and frequently occurring diseases of the participants. The 12 bad childhood experiences included physical abuse, emotional neglect, domestic drug abuse, family mental illness, domestic violence, family members being imprisoned, parents separated or divorced, dangerous neighbors, bullying, death of parents, death of brothers and sisters and disability of parents (http://charls.pku.edu.cn/en/, accessed on 24 September 2020). This objectively creates convenient conditions for assessing the long-term impact of domestic violence on individuals, facilitates tracking the long-term development of China’s population and provides a more scientific basis for formulating and improving China’s relevant policies. It can be said that for China, CHARLS data are the best data to study the impact of domestic violence on individuals. Based on the above reasons, this paper conducts research and analysis based on CHARLS. Based on the existing literature, taking into account the reality of family division of labor, women take on more specific tasks in the process of raising and caring for children, and children’s daily life mainly depends on female caregivers. This paper intends to construct an indicator system for quantifying domestic violence from the four dimensions of injury from violence, negligent care, emotional abuse and witnessing domestic violence (shown in Figure 1). In the life course survey, the respondents recorded in detail whether their parents had beaten them in childhood, whether they had enough experience to take care of themselves, how their relationship with their parents was and whether they had witnessed violence between their parents. The specific definition and quantification of the variables are shown in Table 1.

According to the descriptive statistics, 3.02% of the sample respondents were often beaten by male caregivers, while 4.35% were often beaten by female caregivers, and 6.5% of the respondents were neglected by female caregivers. The proportion of respondents who had bad relationships with male and female caregivers was 1.25 and 0.91%, respectively; 1.75% of respondents’ fathers often beat their mothers, while 0.39% of respondents’ mothers often beat their fathers. Based on the above secondary indicators, combined with the weighting method based on the coefficient of variation method, we estimated the domestic violence index [28]. The secondary indicator and primary indicator weights are also shown in Table 1, and the nuclear density distribution of the domestic violence index is shown in Figure 2. From the distribution of the domestic violence index, the estimation of the kernel density function shows a trailing pattern, and the proportion of respondents experiencing serious domestic violence is relatively low.

### 2.2. The Choice of Variables and the Construction of Empirical Models

#### 2.2.1. Selection of Indicators

This study intends to assess the long-term impact of domestic violence on minors from three aspects: education, health and life satisfaction, so three empirical models need to be built. For the interviewees, aspects such as educational achievements; primary family environment factors, such as parents’ educational level, family economic status, number of siblings, parents’ physical and mental health and whether parents have bad behaviors; demographic variables such as age, gender, nationality, urban or rural area, community environment and economic location; as well as other macro variables are all influencing factors. Among them, the original family environment variables all originate from the 2014 life course survey. The determinants of health are similar to those of educational achievements. In addition to the above factors, education, marriage, family economic conditions and living conditions are also determinants of health [29].

The level of health can be described in two ways: one is through a self-assessment of health; the other is to break up health into physical health and mental health. Physical health can be characterized using biomarker indicators, that is, dimension reduction in blood test indicators. The dimension reduction method is shown in Equation (1) [30]:(1)$$D_{M}\left(x\right)=\sqrt{{\left[x-\mu \left(x\right)\right]}^{T}S^{-1}\left[x-\mu \left(x\right)\right]}$$
where x represents the biomarker indicator vector; μ(x) is its mean vector; and S denotes the covariance matrix of biomarker indicators. Meanwhile, one can also count the frequency of blood test indicators exceeding the threshold value according to the threshold value of each blood test indicator and calculate the risk score. The psychological health risk can be calculated using the test results of the psychological scale. The blood test indicators, their thresholds and the psychological scale are shown in Table 2. The blood examination indicators are from the 2011 and 2015 surveys, while the self-assessment health and psychological surveys have been implemented in four surveys (in the blood test data in 2011, the indicator cystatin C was often missing, so it was not used as an indicator in the dimension reduction in blood test indicators). For life satisfaction, in addition to the above factors, health and education are influencing factors. Education, self-assessment of health, psychological scale, life satisfaction and family living standard indicators are all from the follow-up survey in 2018.

Meanwhile, the life course survey also recorded whether the respondents had often been bullied by other classmates during their school days. Similar to domestic violence, campus bullying can also harm the physical and mental health of minors, so it is necessary to take campus bullying as a control variable. The control variable assignment method is shown in Table 3.

#### 2.2.2. Empirical Model

As variables are exogenous, and education level is an ordered variable, linear model is used for estimation [31]. The empirical model of educational achievement is shown in Equation (2):(2)$$Edu=c_{1}+\alpha_{1}V+\beta_{1}X+\varepsilon_{1}$$
where the control variables X include campus bullying, demographic variables and native family variables. The empirical model of the self-assessment of health and life satisfaction is shown in Equation (3):(3)$$\left\{ \begin{array}{l} SRH_{it}=c_{2}+\alpha_{2}V_{it}+\chi_{2}Edu_{it}+\gamma_{2}Sati_{it}+\beta_{2}X^{\prime }+\varepsilon_{1,it}\\ Sati_{it}=c_{2}+\alpha_{2}V+\chi_{2}Edu_{it}+\gamma_{2}SRH_{it}+\beta_{2}X^{\prime }+\varepsilon_{2,it} \end{array}\right.$$
where the control variables $X^{\prime }$ include campus bullying, demographic statistics, native family variables and variables reflecting the quality of family life. Self-rated health and life satisfaction are both subjective indicators, and there is a causal relationship between them, so they are built into a simultaneous equation model. As self-rated health and life satisfaction are ordered variables, Equation (3) is a bivariate ordered variable model. Health is further divided into two dimensions: physical health and mental health. As physical health and mental health are mutually causal, a simultaneous equation model is also used to quantify the impact of domestic violence on health:(4)$$\left\{ \begin{array}{l} DM_{it}=c_{1}+\alpha_{1}V+\chi_{1}Edu_{it}+\varphi_{1}Depr_{it}+\gamma_{1}Sati_{it}+\beta_{1}X^{\prime }+\varepsilon_{1,it}\\ Depr_{it}=c_{2}+\alpha_{2}V+\chi_{2}Edu_{it}+\phi_{2}DM_{it}+\gamma_{2}Sati_{it}+\beta_{2}X^{\prime }+\varepsilon_{2,it} \end{array}\right.$$
(5)$$\left\{ \begin{array}{l} Risk_{it}=c_{1}+\alpha_{1}V+\chi_{1}Edu_{it}+\varphi_{1}Depr_{it}+\gamma_{1}Sati_{it}+\beta_{1}X^{\prime }+\varepsilon_{1,it}\\ Depr_{it}=c_{2}+\alpha_{2}V+\chi_{2}Edu_{it}+\phi_{2}Risk_{it}+\gamma_{2}Sati_{it}+\beta_{2}X^{\prime }+\varepsilon_{2,it} \end{array}\right.$$

Different from Equation (3), the indicators reflecting physical health (DM), risk scores (Risk) and depression scores (Depr) can be regarded as continuous variables, while life satisfaction is an ordered variable, so Equation (4) is a mixed structure model. In quantitative research, the ordered probit/logit model and the simple linear regression model have consistency in the direction and significance of parameter estimates, with the latter being more intuitive and convenient to explain. Therefore, many studies directly use the OLS estimation ordered choice variable model [32,33], so they can also directly use the seemingly unrelated regression estimator (Equations (2)–(5)).

## 3. Empirical Research

The empirical research includes three main parts: First, the 2018 cross-sectional data are taken as the sample to quantify the impact of domestic violence on personal educational achievements. For the middle-aged and elderly aged 45 and above, the education level was finalized, and the 2018 cross-sectional data can be used as the sample to retain the observation object to the maximum extent. Second, the seemingly unrelated regression model is used to estimate the simultaneous equation of the self-assessment of health and life satisfaction. The sample data are panel data composed of 2011, 2013, 2015 and 2018 survey data. Finally, health is refined into physical health and mental health dimensions, and simultaneous equations are estimated through seemingly unrelated regression. The sample data are panel data composed of 2011 and 2015 survey data.

### 3.1. Domestic Violence and Educational Achievements

Equation (2) is estimated based on sample data. The estimated results are shown in Table 4, which lists the estimated results of the OLS and ordered probit/logit models at the same time. According to the estimation results of the three types of models, at the 1% significance level, domestic violence significantly reduces individual educational achievements. Taking the OLS estimation results as an example, if one unit is added to the domestic violence index, the education level of individuals will decrease by 0.1318 levels. The interpretation of the estimated results of the ordered probit model requires the help of marginal effects. Based on the estimated results of the ordered probit model, the marginal effects of education level on the average value of the domestic violence index $\partial P\left(Edu=\kappa \right)/\partial \bar{V}$ can be estimated, in turn. The estimated results are shown in Figure 3.

It can be seen from the estimation results of the marginal effect that when the domestic violence index takes the average value, the marginal effect of the probability value P(Edu=4) (being educated to graduate from primary school) on the domestic violence index is 0.0056, and for other levels of education, the marginal effect is significantly less than 0. Therefore, it can be seen that domestic violence significantly reduces educational achievements after primary school graduation.

To intuitively explain the estimation results of the ordered logit model, we can also use the generalized ordered logit model in addition to the probability ratio. The generalized ordered logit model converts the ordered logit model into several logit models, which is consistent with the above. Typical primary school graduation, junior high school graduation, senior high school graduation, technical secondary school graduation, junior college graduation and undergraduate graduation are selected as the threshold for model transformation; that is, the impact of the domestic violence index on the probability value P(Edu≥k|X)(k=4,5,⋯,9) is mainly examined, with the estimation results of the probability ratio shown in Figure 3. It can be seen from the estimated results of the probability ratio that, if the domestic violence index increases by 1 unit, the probability ratio of attaining primary school graduation and above will decrease by 13.42%, the probability ratio of attaining junior high school graduation and above will decrease by 13.72% and the probability ratios of attaining high school graduation, technical secondary school graduation, junior college graduation, undergraduate graduation and above will decrease by 21.11, 16.94, 14.45 and 17.61%, respectively. According to the estimation results of the OLS estimation, the ordered probit/logit model and the generalized logit model, domestic violence significantly reduces the educational achievements of respondents.

The domestic violence index is composed of four dimensions, and the impact of each dimension on educational achievements may be inconsistent. In view of this, in the heterogeneity analysis, the domestic violence index is subdivided into four dimensions, and the corresponding estimation results are shown in Table 5. It can be seen from the above estimation results that the OLS estimation and the coefficient estimation of the ordered probit/logit model are consistent in significance and sign, so the OLS estimation results of the linear model are used to explain the practical meaning of the model. At the 1% confidence level, among the four dimensions, only the emotional abuse dimension has a significant negative impact on educational achievement; that is, compared with the other three dimensions, emotional abuse has the most prominent negative impact on educational achievement. Specifically, if the emotional abuse index increased by 1 unit, the education level decreased by 0.0759. This is because emotional abuse will affect children’s cognitive development and impair their memory and cognitive ability to a certain extent, making them likely to encounter difficulties in learning, thus affecting their academic performance and then their education level. From another perspective, scholars have found that the level of education will adjust the impact of domestic violence on individuals, so the level of education is an important factor to consider the impact of domestic violence on individuals [34].

### 3.2. Domestic Violence, Health and Life Satisfaction

Similar to the above, this part also uses the linear model for empirical research. The Breusch–Pagan test shows that the residual terms of the simultaneous equations are correlated, so the seemingly uncorrelated panel model is used to estimate the simultaneous equations. The estimation results are shown in Table 6. At the 1% confidence level, the domestic violence index has a significant negative impact on the self-assessment health level and life satisfaction. If the domestic violence index increases by 1 unit, the self-assessment health level decreases by 0.0320, and life satisfaction decreases by 0.0948. Furthermore, the domestic violence index is divided into four levels. For health self-evaluation, at the 1% confidence level, only the emotional abuse dimension has a significant negative impact on the health self-evaluation level, which increases by 1 unit, while the self-evaluation health level decreases by 0.0267. In the life satisfaction equation, at the 1 or 5% confidence level, injury from violence, negligent care, emotional abuse and witnessing domestic violence all have significant negative impacts on life satisfaction. For each increase in the index of each dimension, life satisfaction decreases by 0.0240, 0.0189, 0.0314, and 0.0216 levels, in turn. In general, domestic violence significantly reduces the self-rated health level and life satisfaction. This is because domestic violence causes great harm to the victims, directly damages the physical and mental health of the victims and causes long-term mental tension, anxiety and fear in the victims. At the same time, because domestic violence makes it difficult for victims to feel warmth from family, life satisfaction will be greatly reduced.

### 3.3. Further Discussion on Domestic Violence and Health

On the basis of the above, health is further divided into physical health and mental health, characterized by biomarker indicators and depression score indicators. The corresponding estimation results are shown in Table 7. At the 1% confidence level, the domestic violence index has a significant positive impact on depression scores; at the 5% confidence level, the domestic violence index significantly increases the abnormal frequency of blood test indicators. Specifically, in the simultaneous equation of DM and depression scores, if the domestic violence index increased by 1 unit, the depression score increased by 0.6591 points; in the simultaneous equation of the abnormal frequency of blood test index and depression scores, if the domestic violence index increased by 1 unit, the abnormal frequency of blood test index increased by 0.0532 units, and the depression score increased by 0.6617 points. Furthermore, the domestic violence index is divided into four dimensions. At the 1% confidence level, the three indexes of injury from violence, emotional abuse and witnessing domestic violence significantly improved the depression score but have no significant impact on the two health risk indicators based on blood test indicators. Therefore, on the whole, it can be determined that domestic violence increases the subjective mental health risk.

### 3.4. Robustness Test

Calculating the domestic violence index through dimension reduction can quantify the degree of domestic violence experienced by the interviewees in general, but it will also lose some of the indicator information. In view of this, in the robustness test, directly using the secondary indicators as explanatory variables is proposed, with the estimated results shown in Table 8. In the education decision equation, at the 1% confidence level, only the relationship with the mother has a significant negative impact on education level. In the simultaneous equation of self-rated health and life satisfaction, for self-rated health, at the 5% confidence level, only the relationship with the mother has a significant negative impact. For life satisfaction, at the 1% confidence level, whether the father has injuries from violence, whether the mother has invested enough in taking care of herself and the relationship with the father have significant negative effects. In the two simultaneous equations of health risk, seven secondary indicators have no significant impact on the health risk indicators based on blood test indicators. For subjective mental health, at the 1 or 5% confidence level, whether the mother behaved violently, the relationship with the mother and whether domestic violence was witnessed have significant positive effects on the depression score. In general, the secondary indicators in the dimension of emotional abuse have a particularly prominent impact on educational achievement, life satisfaction and mental health, which verifies the main conclusions of the empirical study.

## 4. Conclusions and Policy Recommendations

Domestic violence includes not only physical violence but also mental violence with regard to neglect, emotional abuse, etc. Therefore, this study estimates a domestic violence index from the four aspects of injury from violence, negligent care, emotional abuse and witnessing domestic violence, and then takes the CHARLS (2011, 2013, 2015, 2018) and the “life course” survey as sample data to assess the impact of domestic violence on personal education, health and life satisfaction, in turn. The main conclusions are as follows: (1) Domestic violence significantly reduced the respondents’ educational achievements. Compared with the three dimensions of injury from violence, negligent care and witnessing domestic violence, emotional abuse had the most significant negative impact on educational achievements. (2) Domestic violence significantly reduced the self-rated health level and life satisfaction and significantly increased the mental health risk of the respondents.

The above conclusions have important policy implications for optimizing social governance strategies. Domestic violence has far-reaching negative impacts on personal education, health and life satisfaction. To prevent domestic violence and heal the trauma caused, based on its complexity and concealment, we believe that its long-term impact on individuals should be approached from the following four perspectives.

First, a domestic violence monitoring system should be built. Domestic violence has the characteristics of being long-term and repeated, so it is necessary to find the families involved and prevent recurrence in a timely manner. On one hand, the tracking mechanism should be strengthened: for people with low educational achievements and low physical and mental satisfaction (especially young people), society, schools and families should be vigilant in tracing domestic violence back to the source to prevent long-term negative impacts. On the other hand, the feedback mechanism should be strengthened: for those who have suffered from domestic violence, the probability of being subjected to repeated domestic violence is greatly increased. Therefore, they should be encouraged to express their concerns freely, and in the future, a “one-to-one” follow-up mechanism, and a “fixed + random” feedback mechanism should be established to strengthen the ability of victims to provide feedback and communicate with the relevant departments.

Second, the harm caused by emotional abuse and other mental abuse should be confronted. On one hand, the consciousness of the victims needs to be awakened. Domestic violence refers not only to physical violence but also emotional abuse, neglect and other spiritual mistreatment. However, compared with physical violence, the biggest dilemma surrounding domestic psychological abuse is that the victims do not comprehend it themselves but instead feel extreme emotional pain and depression. Therefore, it is necessary to make the content and methods of domestic psychological abuse known, so that the parties who are unknowingly experiencing it will become aware and safeguard their rights. On the other hand, we should establish a working mechanism for linking the authorities that deal with domestic violence. The difficulty in determining if domestic violence is occurring is that it is not easy to obtain evidence, and many victims are unable to enter the judicial process. Therefore, the judicial department should link with women’s federations, neighborhood committees, village committees and other departments to deal with cases of psychological abuse flexibly and quickly, integrating evidence collection, assistance and protection.

Third, attention should be paid to the long-term impact of domestic violence on individuals. On one hand, many perpetrators do not realize that domestic violence is a crime; on the other hand, they ignore the long-term harm to individuals caused by domestic violence. Therefore, we should not only enhance the public’s legal understanding of domestic violence but also use new media to publicize the serious harm that can be caused to individuals as a result of domestic violence. Furthermore, family moral education needs to be strengthened, and the establishment of harmonious families advocated.

Fourth, it is necessary for domestic violence to be prevented at the source. Accordingly, we must go deep into communities to facilitate an understanding of the legal issues related to family disputes [35,36], not only to issue personal safety protection orders to the victims but also to use laws and regulations to intervene and correct the behavior of the perpetrators [37]. Finally, we need to fully investigate and establish a family violence litigation protection base and form a “one-stop” litigation processing procedure that is simple and smooth, with privacy protections.

## Figures and Tables

**Figure 1 behavsci-13-00137-f001:**
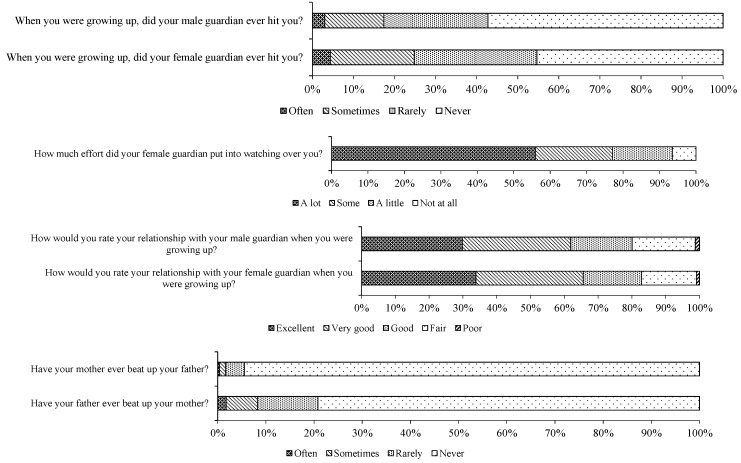
Descriptive statistics of domestic violence dimensions.

**Figure 2 behavsci-13-00137-f002:**
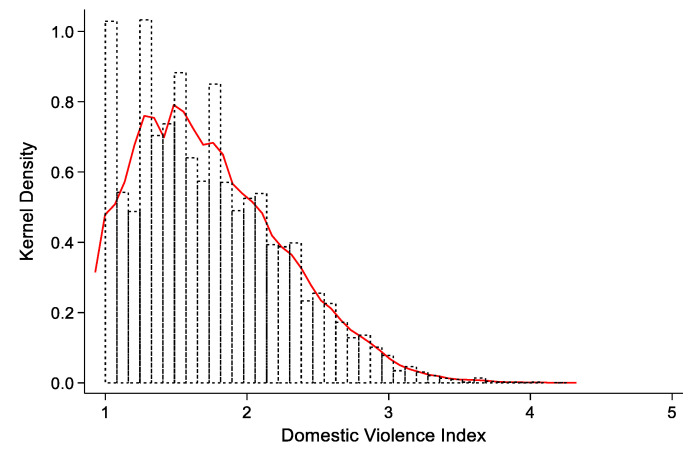
Nuclear density estimation of domestic violence index.

**Figure 3 behavsci-13-00137-f003:**
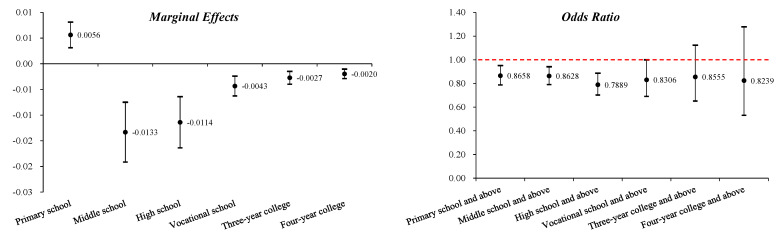
Marginal effect and probability ratio of education level on domestic violence index.

**Table 1 behavsci-13-00137-t001:** Domestic violence dimensions and quantitative methods.

Dimension	Symbol	Quantification Method	Weight	
			Secondary Indicators	Level I Indicators
Injury from violence	*V* _11_	Did your female caregivers beat you when you were young? 1. Never; 2. Rarely; 3. Sometimes; 4. Often	0.4869	0.2334
	*V* _12_	When you were a child, did your male caregiver beat you? 1. Never; 2. Rarely; 3. Sometimes; 4. Often	0.5131	
Negligent care	*V* _2_	When you were young, did your female caregivers spend a lot of energy taking care of you? 1. A lot; 2. Some; 3. A little; 4. Not at all	1	0.2986
Emotional abuse	*V* _31_	How do you evaluate your relationship with your female caregivers when you were young? 1. Poor; 2. Fair; 3. Good; 4. Very good; 5. Excellent	0.5057	0.2530
	*V* _32_	How would you rate your relationship with your male dependants when you were young? 1. Poor; 2. Fair; 3. Good; 4. Very good; 5. Excellent	0.4943	
Witness to domestic violence	*V* _41_	Has your father ever beaten your mother? 1. Never; 2. Rarely; 3. Sometimes; 4. Often	0.6168	0.2150
	*V* _42_	Has your mother ever beaten your father? 1. Never; 2. Rarely; 3. Sometimes; 4. Often	0.3832	

**Table 2 behavsci-13-00137-t002:** Blood test indicators and psychological scale.

Serial No.	Blood Test Index			Psychological Scale
	Index Name (English)	Index Name (Chinese)	Threshold	
1	White blood cell (in thousands)	White blood cell	–	I was annoyed by some trifles
2	Hemoglobin (g/dL)	Hemoglobin	Male: <13 g/dL Female: <12 g/dL	It’s hard for me to concentrate on things
3	Hematocrit (%)	Hematocrit	–	I feel down
4	Mean corpuscular volume (fl)	Mean corpuscular volume	–	I find it hard to do anything
5	Platelets (109/L)	Platelet	–	I am full of hope for the future (reverse processing)
6	Triglycerides (mg/dL)	Triglyceride	≥200 mg/dL	I feel scared
7	Creatinine (mg/dL)	Creatinine	>1.4 mg/dL	I don’t sleep well
8	Blood urea nitrogen (mg/dL)	Blood urea nitrogen	>20 mg/dL	I’m very happy (reverse processing)
9	HDL cholesterol (mg/dL)	High-density lipoprotein cholesterol	<40 mg/dL	I feel lonely
10	LDL cholesterol (mg/dL)	Low-density lipoprotein cholesterol	>160 mg/dL	I feel I can’t go on with my life
11	Total cholesterol (mg/dL)	Total cholesterol	≥240 mg/dL	
12	Glucose (mg/dL)	Glucose	≥126 mg/dL	
13	Uric acid (mg/dL)	Uric acid	–	
14	C-reactive protein (mg/L)	C-reactive protein	>3 mg/L	
15	Glycated hemoglobin (%)	Glycosylated hemoglobin	≥6.5%	

**Table 3 behavsci-13-00137-t003:** Interpreted, explanatory and control variables.

Category	Variable		Symbol	Definition
Interpreted variables	Education		*Edu*	1. Uneducated (illiterate), 2. Uneducated primary school, 3. Private school, 4. Primary school, 5. Junior high school, 6. High school, 7. Technical secondary school (including secondary normal school and vocational high school), 8. Junior college, 9. Undergraduate, 10. Master, 11. Doctor
	Healthy		Self-rated health: SRH Blood test index: DM Risk score: Risk Depression score: Depr	Self-rated health: 1. Very bad, 2. Bad, 3. Average, 4. Good, 5. Very good; Blood test indicators: reduce the dimension of blood test indicators through Markov distance function; Risk score: judge whether the blood test indicators are normal according to the threshold, and add up the number of abnormal indicators; Depression score: calculated using depression scale
	Life satisfaction		*Sati*	1. Not at all satisfied, 2. Not very satisfied, 3. Quite satisfied, 4. Very satisfied, 5. Extremely satisfied
Explanatory variable	Domestic violence index		*V*	The score is calculated from the four dimensions of injury from violence, negligent care, emotional abuse and witnessing domestic violence
Control variable	Campus bullying		*Bullying*	When you were young, were you bullied by other students at school? 1. Never, 2. Rarely, 3. Sometimes, 4. Often
	Demography Statistics variable	Age	*Age*	Age of the interviewee
		Gender	*Gender*	1: Male, 0: female
		Registered residence	*Hukou*	Respondent’s first household registration: 1: non-agricultural household registration, 0: agricultural household registration
		Nation	*Minzu*	1: Han nationality, 0: others
		Marriage	*Marr*	1: Married, 0: unmarried
		Region	*West*, central, east	The economic region of the interviewee—west: west, central: central, east: east
	Aboriginal family Court variable	Father’s education level	*Edu_f_*	Consistent with the definition of respondents’ education level
		Education level of mother	*Edu_m_*	Consistent with the definition of respondents’ education level
		Father’s mental health	*Depr_f_*	Has your male caregiver ever been sad or depressed for two or more consecutive weeks? 1: Yes, 0: no
		Mother’s mental health	*Depr_m_*	Has your female caregiver ever been sad or depressed for two or more consecutive weeks? 1: Yes, 0: no
		Father’s health	*Sick_f_*	Does your male caregiver stay in bed for a long time? 1: Yes, 0: no
		Mother’s health	*Sick_m_*	Does your female caregiver stay in bed for a long time? 1: Yes, 0: No
		Number of brothers and sisters	*Siblings*	Number of brothers and sisters in the family
		Does my father drink too much?	*Alcoh*	1: Yes, 0: no
		Whether the father takes drugs	*Drug*	1: Yes, 0: no
		Does my father gamble?	*Gambling*	1: Yes, 0: no
		Family economic status	*Situ*	Before the age of 17, compared with the ordinary families in your community/village at that time, what was your family’s economic situation? 1. A lot worse than them, 2. A little worse than them, 3. The same as them, 4. A little better than them, 5. A lot better than them
		Community health	*Comm*	1. Not clean and tidy at all, 2. Not very clean and tidy, 3. Quite clean and tidy, 4. Very clean and tidy
	Family Students Live mass	Living standard	*Durables*	Quantity of 18 kinds of household equipment, durable consumer goods and other valuables
		Toilet	*Toilets*	1: There is a flush toilet at home, 0: no
		Tap water	*Water*	1: There is tap water at home, 0: no
		Fuel	*Fuel*	1: The main fuels for cooking are straw and firewood, 0: others
		Internet	*Internet*	1: The house you live in can have broadband internet access, 0: no

**Table 4 behavsci-13-00137-t004:** Interpreted, explanatory and control variables.

	(1)	(2)	(3)
	*OLS*	Ordered Probit Model	Ordered Logit Model
Variable	*edu*	*edu*	*edu*
*V*	−0.1318 ***	−0.0951 ***	−0.1557 ***
	(0.0307)	(0.0212)	(0.0368)
*Bullying*	−0.0099	−0.0047	−0.0216
	(0.0232)	(0.0160)	(0.0281)
*Age*	−0.0178 ***	−0.0129 ***	−0.0250 ***
	(0.0018)	(0.0013)	(0.0021)
*Gender*	0.6911 ***	0.4814 ***	0.8277 ***
	(0.0316)	(0.0221)	(0.0386)
*Hukou*	−0.9048 ***	−0.6448 ***	−1.1659 ***
	(0.0493)	(0.0345)	(0.0617)
*Minzu*	0.0938	0.0613	0.1156
	(0.0634)	(0.0436)	(0.0779)
*Central*	0.2589 ***	0.1931 ***	0.3401 ***
	(0.0389)	(0.0268)	(0.0467)
*East*	0.1881 ***	0.1360 ***	0.2413 ***
	(0.0384)	(0.0264)	(0.0460)
*Edu_f_*	0.1564 ***	0.1075 ***	0.1845 ***
	(0.0156)	(0.0106)	(0.0187)
*Edu_m_*	0.1206 ***	0.0842 ***	0.1457 ***
	(0.0103)	(0.0071)	(0.0124)
*Depr_f_*	−0.1351 **	−0.0864 **	−0.1531 **
	(0.0634)	(0.0436)	(0.0756)
*Depr_m_*	−0.1969 ***	−0.1342 ***	−0.2421 ***
	(0.0595)	(0.0412)	(0.0724)
*Sick_f_*	−0.0860 *	−0.0508	−0.1063 *
	(0.0516)	(0.0353)	(0.0617)
*Sick_m_*	−0.0928 *	−0.0721 **	−0.1287 **
	(0.0484)	(0.0336)	(0.0577)
*Siblings*	−0.0113	−0.0050	−0.0068
	(0.0088)	(0.0060)	(0.0105)
*Alcoh*	−0.1007 *	−0.0577	−0.1071
	(0.0593)	(0.0405)	(0.0700)
*Drug*	0.0298	−0.0391	−0.1376
	(0.3549)	(0.2225)	(0.3348)
*Gambling*	−0.2785 **	−0.2048 **	−0.4052 ***
	(0.1314)	(0.0897)	(0.1475)
*Situ*	0.2301 ***	0.1631 ***	0.2854 ***
	(0.0172)	(0.0119)	(0.0208)
*Comm*	−0.0230	−0.0203	−0.0303
	(0.0209)	(0.0144)	(0.0250)
Constant term	4.6869 ***		
	(0.1794)		
Observation object	9642	9642	9642
*R* ^2^	0.1808		

Note: Robust standard deviation in brackets; *** *p* < 0.01, ** *p* < 0.05, * *p* < 0.1; the estimated result of the tangent point value is omitted.

**Table 5 behavsci-13-00137-t005:** Results of the dimensional heterogeneity analysis.

Variable	(1)	(2)	(3)
	*OLS*	Ordered Probit Model	Ordered Logit Model
	*Edu*	*Edu*	*Edu*
*V* _1_	0.0288	0.0238	0.0506 *
	(0.0229)	(0.0158)	(0.0277)
*V* _2_	−0.0228	−0.0159	−0.0254
	(0.0178)	(0.0122)	(0.0212)
*V* _3_	−0.0759 ***	−0.0565 ***	−0.0970 ***
	(0.0162)	(0.0112)	(0.0193)
*V* _4_	−0.0283	−0.0217	−0.0399
	(0.0347)	(0.0240)	(0.0419)
Constant term	4.6315 ***		
	(0.1818)		
Observation object	9642	9642	9642
*R* ^2^	0.1819		
Control variable	√	√	√

Note: Robust standard deviation in brackets; *** *p* < 0.01, * *p* < 0.1; the estimated results of control variables and tangent point values are omitted.

**Table 6 behavsci-13-00137-t006:** Estimated results of domestic violence, health and life satisfaction.

	(1)	(2)	(3)	(4)
Variable	*SRH*	*Sati*	*SRH*	*Sati*
*Sati*	0.5700 ***		0.5681 ***	
	(0.0079)		(0.0079)	
*SRH*		0.3444 ***		0.3434 ***
		(0.0048)		(0.0048)
*V*	−0.0320 ***	−0.0948 ***		
	(0.0116)	(0.0090)		
*V* _1_			−0.0005	−0.0240 ***
			(0.0087)	(0.0067)
*V* _2_			0.0078	−0.0189 ***
			(0.0066)	(0.0051)
*V* _3_			−0.0276 ***	−0.0314 ***
			(0.0061)	(0.0048)
*V* _4_			−0.0050	−0.0216 **
			(0.0129)	(0.0100)
Constant term	1.5371 ***	1.6178 ***	1.5406 ***	1.6256 ***
	(0.0755)	(0.0582)	(0.0760)	(0.0586)
Sample size	23,861	23,861	23,861	23,861
*R* ^2^	0.0903	0.0779	0.0914	0.0786
Control variable	√	√	√	√
Time-fixed effect	√	√	√	√

Note: Robust standard deviation in brackets; *** *p* < 0.01, ** *p* < 0.05; the estimated results of other control variables and tangent point values are omitted.

**Table 7 behavsci-13-00137-t007:** Estimated results of domestic violence and physical and mental health.

	(1)	(2)	(3)	(4)	(5)	(6)	(7)	(8)
	Simultaneous Equation (1)		Simultaneous Equation (2)		Simultaneous Equation (3)		Simultaneous Equation (4)	
Variable	*DM*	*Depr*	*Risk*	*Depr*	*DM*	*Depr*	*Risk*	*Depr*
*Depr*	0.0147 ***		0.0063 ***		0.0148 ***		0.0061 **	
	(0.0029)		(0.0024)		(0.0029)		(0.0024)	
*DM*		0.2004 ***				0.2014 ***		
		(0.0395)				(0.0395)		
*Risk*				0.1291 ***				0.1252 **
				(0.0487)				(0.0486)
*V*	0.0400	0.6591 ***	0.0532 **	0.6617 ***				
	(0.0329)	(0.1210)	(0.0267)	(0.1210)				
*V* _1_					0.0184	0.2255 **	0.0265	0.2264 **
					(0.0249)	(0.0918)	(0.0203)	(0.0919)
*V* _2_					0.0240	0.0562	0.0192	0.0588
					(0.0186)	(0.0687)	(0.0151)	(0.0687)
*V* _3_					−0.0018	0.1838 ***	−0.0031	0.1842 ***
					(0.0174)	(0.0640)	(0.0141)	(0.0640)
*V* _4_					−0.0054	0.4847 ***	0.0428	0.4793 ***
					(0.0366)	(0.1348)	(0.0298)	(0.1349)
Constant term	2.6178 ***	18.4491 ***	0.0666	19.0057 ***	2.6204 ***	18.1320 ***	0.0401	18.6963 ***
	(0.2124)	(0.7650)	(0.1726)	(0.7566)	(0.2139)	(0.7710)	(0.1738)	(0.7627)
Sample size	8698	8698	8698	8698	8698	8698	8698	8698
*R* ^2^	0.0200	0.1255	0.0162	0.1255	0.0201	0.1267	0.0165	0.1267
Control variable	√	√	√	√	√	√	√	√
Time-fixed effect	√	√	√	√	√	√	√	√

Note: Robust standard deviation in brackets; *** *p* < 0.01, ** *p* < 0.05; the estimated results of other control variables and tangent point values are omitted.

**Table 8 behavsci-13-00137-t008:** Estimation results of the robustness test.

Variable	(1)	(2)	(3)	(4)	(5)	(6)	(7)
	Equation (2)	Equation (3)		Equation (4)		Equation (5)	
	*edu*	*SRH*	*Sati*	*DM*	*Depr*	*Risk*	*Depr*
*V* _11_	0.0271	−0.0071	−0.0065	−0.0119	0.1874 **	0.0105	0.1841 **
	(0.0203)	(0.0076)	(0.0059)	(0.0216)	(0.0797)	(0.0176)	(0.0797)
*V* _12_	−0.0009	0.0079	−0.0184 ***	0.0342	0.0179	0.0170	0.0228
	(0.0227)	(0.0086)	(0.0067)	(0.0247)	(0.0909)	(0.0201)	(0.0909)
*V* _2_	−0.0189	0.0083	−0.0198 ***	0.0247	0.0391	0.0190	0.0418
	(0.0179)	(0.0066)	(0.0052)	(0.0188)	(0.0693)	(0.0153)	(0.0693)
*V* _31_	−0.0825 ***	−0.0178 **	−0.0086	0.0006	0.2280 ***	0.0002	0.2286 ***
	(0.0225)	(0.0083)	(0.0064)	(0.0235)	(0.0867)	(0.0191)	(0.0867)
*V* _32_	0.0051	−0.0100	−0.0227 ***	−0.0023	−0.0394	−0.0034	−0.0395
	(0.0219)	(0.0080)	(0.0062)	(0.0229)	(0.0843)	(0.0186)	(0.0843)
*V* _41_	−0.0114	−0.0129	−0.0099	−0.0132	0.4454 ***	0.0141	0.4419 ***
	(0.0272)	(0.0101)	(0.0079)	(0.0288)	(0.1061)	(0.0234)	(0.1061)
*V* _42_	−0.0366	0.0218	−0.0124	0.0154	−0.1544	0.0529	−0.1583
	(0.0490)	(0.0187)	(0.0145)	(0.0539)	(0.1986)	(0.0438)	(0.1987)
*Sati*		0.5682 ***					
		(0.0079)					
*SRH*			0.3434 ***				
			(0.0048)				
*Depr*				0.0149 ***		0.0061 ***	
				(0.0029)		(0.0024)	
*DM*					0.2027 ***		
					(0.0395)		
*Risk*							0.1265 ***
							(0.0486)
Constant term	4.6471 ***	1.5254 ***	1.6271 ***	2.6113 ***	18.3200 ***	0.0161	18.8899 ***
	(0.1851)	(0.0770)	(0.0594)	(0.2166)	(0.7802)	(0.1760)	(0.7720)
Sample size	9642	23,861	23,861	8698	8698	8698	8698
*R* ^2^	0.1824	0.0915	0.0787	0.0202	0.1275	0.0166	0.1275
Control variable	√	√	√	√	√	√	√
Fixed-time effect	√	√	√	√	√	√	√

Note: Robust standard deviation in brackets; *** *p* < 0.01, ** *p* < 0.05; the estimated results of other control variables and tangent point values are omitted.

## Data Availability

This is not applicable to this article as no datasets were generated.

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
