# Peer review of "Long-Term Impact of Domestic Violence on Individuals—An Empirical Study Based on Education, Health and Life Satisfaction"

_behavsci, 2023, doi:10.3390/bs13020137_

Round 1

Reviewer 1 Report (New Reviewer)

The article is well written, deals with an interesting topic and offers important indications.

I have only a few suggestions for authors:

The abstract lacks information about the composition of the sample involved in the research and the type of analysis used.

The article dwells a lot on the technical and analytical aspects (which are impeccable), but the discussion part of the results found seems a bit scarce to me. The authors should further investigate the implications of the results found and compare these results with those present in the literature.

Finally, in the conclusions, the authors could specify whether the strategies identified to prevent domestic violence are in line with other approaches suggested by the international literature.

See for example:

Semahegn, A., Torpey, K., Manu, A., Assefa, N., Tesfaye, G., & Ankomah, A. (2019). Are interventions focused on gender-norms effective in preventing domestic violence against women in low and lower-middle income countries? A systematic review and meta-analysis. Reproductive health, 16(1), 1-31.

Esposito, C., Di Napoli, I., Esposito, C., Carnevale, S., & Arcidiacono, C. (2020). Violence against women: a not in my back yard (NIMBY) phenomenon. Violence and gender, 7(4), 150-157.

Oliver, W. (2000). Preventing domestic violence in the African American community: The rationale for popular culture interventions. Violence Against Women, 6(5), 533-549.

Michau, L., & Voices, R. (2005). Good practice in designing a community-based approach to prevent domestic violence. In Paper at the Expert Group Meeting Workshop ‘Violence Against women: Good Practices in Combating and Eliminating Violence Against Women’, Organized by UN Division for the Advancement of Women.

Author Response

Dear reviewers and editors:

      Thank you very much for your comments and guidance on our paper. Thank you for pointing out that “The article is well written, deals with an interesting topic and offers important indications” and “The article wells a lot on the technical and analytical aspects (which are impossible)”, which is a great encouragement to us. In response to your guidance, we have made the following modifications:

Point 1:The abstract lacks information about the composition of the sample involved in the research and the type of analysis used.

Answer 1: Thank you for your guidance. In the summary section, we have added information about the sample composition and the type of analysis used (page 1,in red).

Point 2:The article dwells a lot on the technical and analytical aspects (which are impeccable), but the discussion part of the results found seems a bit scarce to me. The authors should further investigate the implications of the results found and compare these results with those present in the literature.

Answer 2: Thank you very much for your shortcomings in the results. We have added the discussion part of the results (page 11,in red; page 12,in red).

Point 3:Finally, in the conclusions, the authors could specify whether the strategies identified to prevent domestic violence are in line with other approaches suggested by the international literature.

Answer 3: Thank you very much for listing the very valuable references for us. We have added these documents to the paper, especially the conclusion part, which makes the paper more convincing and scientific(page 15,16,in red).

    I sincerely thank you for your professional and rigorous guidance. Under your careful, objective and professional guidance, the quality of the thesis has been further improved. If there are any deficiencies, I hope you can continue to guide us and pay deep respect to all the work you have done for our paper.

Sincerely,

Yating Peng

Hunan Agricultural University

[email protected]

January 25, 2023

Reviewer 2 Report (New Reviewer)

I thank the authors for the opportunity to read and evaluate the text. This is a relevant and original topic, of interest to different regions of the globe.

The stated objective is: "Based on the existing literature, this paper aims to construct a system of indicators to quantify domestic violence from the four dimensions of violence: injury, neglectful care, emotional abuse, and witnessing domestic violence."

Although the statistical organization and analysis of the data seems correct to me, I request that they be reviewed, according to the journal's recommendation, by a statistical expert. Once the statistical analysis is confirmed, the results are relevant and meet what was formulated in the goals of the study.

My suggestion concerns one element of Table 1, it is admirable that the item on neglect is measured only in relation to the adult female member: "Neglectful care V2 When you were young, did your caregivers spend a lot of energy caring for you? 1. a lot; 2. some; 3. a little; 4. not at all". In this regard, it would be important to add considerations about the country's expectation of the roles of men and women in the upbringing and development of children and adolescents. This will help advance the debate on the subject and the formative policies for family members.

Sincerely.

Author Response

Dear reviewers and editors:

Thank you very much for your comments and guidance on our paper. Thank you for pointing out that “I thank the authors for the opportunity to read and evaluate the text. This is a relative and original topic, of interest to different regions of the globe”, which is a great encouragement to us. In response to your guidance, we have made the following modifications:

Point 1:Although the statistical organization and analysis of the data seems correct to me, I request that they be reviewed, according to the journal’s recommendation, by a statistical expert. Once the statistical analysis is confirmed, the results are relevant and meet what was formulated in the goals of the study.

Answer 1: Thank you very much for your comments and guidance. The author of the paper has been committed to statistical research for a long time, has received good professional training in statistics and econometrics, has a lot of experience in building an indicator system and conducting quantitative research, can skillfully use micro-investigation data for research, and is familiar with the latest progress of research in this field. We can assure you that the data processing in this paper is in compliance with the specifications, and the model processing and statistical estimation results are very reliable.

Point 2:My suggestion concerns one element of Table 1, it is admirable that the item on neglect is measured only in relation to the adult female member: “Neglectful care V2 When you were young, did your caregivers spend a lot of energy caring for you? 1. a lot; 2. some; 3. a little; 4. not at all”. In this regard, it would be important to add considerations about the country’s expectation of the roles of men and women in the upbringing and development of children and adolescents. This will help advance the debate on the subject and the formative policies for family members.

Answer 2: Thank you very much for your guidance on Table 1. First of all, please allow us to explain the following reasons to you. Table 1 is set for women because of the reality of China. In the social division of labor in China, women take on more responsibilities in the process of raising and caring for children. Children’s daily life mainly depends on women’s caregivers. Secondly, under your guidance, we added the explanation of why V2 is set as female in the paper, which makes the paper more convincing and complete (page 3, in red).

In addition, we have increased the discussion of the results and added relevant references to make the conclusion more convincing(page 1,11,12,15,16,in red). I sincerely thank you for your professional and rigorous guidance. Under your careful, objective and professional guidance, the quality of the thesis has been further improved. If there are any deficiencies, I hope you can continue to guide us and pay deep respect to all the work you have done for our paper.

Sincerely,

Yating Peng

Hunan Agricultural University

[email protected]

January 25, 2023

This manuscript is a resubmission of an earlier submission. The following is a list of the peer review reports and author responses from that submission.

Round 1

Reviewer 1 Report

The authors submit a paper that aims to assess the long-term impacts of domestic violence experienced in childhood on individuals through a survey of health and elderly care in China. They conclude in general that an experience of domestic violence reduces educational achievements, and that domestic violence reduces the self-assessed health level and life satisfaction, increasing the subjective mental health risk.

However, the sample and participants' data are not presented. There is an argumentative strategy that doesn´t clearly define the previous literature about the main relationship between the variables of the study. Instead, the authors decide to include information about laws and international-convention guidelines that make theoretical and methodological justifications unclear.

Moreover, the sections of the paper don´t follow the traditional description of antecedents, methods, results, discussion, and limitations. It makes it difficult to understand the argumentative strategy about the validity of the used instruments of indirect measurement or the sample characteristics that justify the kind of statistical analysis. Therefore, it is hard to understand the design and statistical management of the data to accomplish the goal of the study.

Furthermore, there is no previous evidence about the models proposed and the statistic used. These difficulties lead to a non-existent internal consistency in the study. Authors should describe the size of the sample, the inclusion-exclusion criteria while considering the data, and the basis of the kind of statistical models proposed. Might be helpful to describe the relationships between domestic violence and education, health, and life satisfaction based on the traditional adjustment indexes of the proposed models. I hope the recommendations might lead to a submit a clear paper to the journal.

Thank you very much for the opportunity of contributing to this paper. 

Author Response

Dear editor,

Thank you very much for your valuable comments. We have benefited a lot and made a lot of changes.

First of all, please allow me to explain the China Health and Retirement Longitudinal Survey (CHARLS) to you about your question about the use of data. First, for China, CHARLS is the best data to study the impact of domestic violence on individuals, creating good conditions for assessing the long-term impact of domestic violence on individuals. CHARLS is a large interdisciplinary survey project hosted by the National Development Research Institute of Peking University and implemented by the China Social Science Investigation Center. It is a major project funded by the National Natural Science Foundation of China. CHARLS survey was conducted in 2011, 2013, 2015 and 2018. The sample covers 150 counties, 450 communities (villages) and 12400 households in 28 provinces (autonomous regions, municipalities directly under the Central Government) across the country. There are 19000 respondents, with a large sample size. In particular, CHARLS conducted a detailed survey on whether the interviewees suffered from domestic violence and bullying in their childhood, and collected information on 12 bad childhood experiences, 14 chronic diseases and frequently occurring diseases of the participants. The 12 bad childhood experiences included physical abuse, emotional neglect, domestic drug abuse, family mental illness, domestic violence, family members being imprisoned, parents separated or divorced, dangerous neighbors, bullying Death of parents, death of brothers and sisters and disability of parents(http://charls.pku.edu.cn/en/,accessed on 24 September 2020). This objectively creates convenient conditions for assessing the long-term impact of domestic violence on individuals, facilitates tracking the long-term population development in China, and provides a more scientific basis for this study; Second, CHARLS is the authoritative data at home and abroad, with a high degree of use. On August 23, 2013, Science published a comprehensive special report on CHARLS. In 2015, CHARLS completed the second regular tracking survey of national baseline samples. This year, the number of CHARLS data users was close to 10000, and 440 academic papers based on CHARLS data have been published at home and abroad. The National Natural Science Foundation of China highly praised CHARLS’ achievements and contributions in basic data construction during the mid-term assessment. In 2017, CHARLS, as a supplement to the provincial representative sample and a preliminary study of the baseline questionnaire, had more than 20000 users of CHARLS data, 708 academic papers based on CHARLS data have been published at home and abroad, and 18890 papers based on CHARLS data have been published at home and abroad in 2022. As a result, this paper adds a detailed description of CHARLS data, as well as references, which provides a solid foundation for the following analysis (see page 3).

Secondly, after your kind reminder, we have added a description of the model and references (see page 8). Since the variables are exogenous, the level of education is an ordered variable, and self-assessment health and life satisfaction are subjective indicators, and there is a causal relationship between the two, so the two are built into a simultaneous equation model. Since both self rated health and life satisfaction are ordered variables, a bivariate ordered variable model is used. Health is further divided into two dimensions: physical health and mental health. Because physical health and mental health are mutually causal, simultaneous equation model is also used to quantify the impact of domestic violence on health.

Finally, please allow me to explain my actual situation. This is a critical period for my doctoral graduation. We clearly require to publish an influential paper after graduation, and your journal is a very influential journal in the industry. My tutor has always asked me to make constant efforts to achieve the goal of publishing in your journal. So we have done a lot of work in the early stage, and we have also carried out professional polishing services and layout in your journal in the later stage, in order to better meet your requirements for papers.

Thank you very much for your suggestions on modification. I don’t want to give up the goal I have always set. I really hope you can give us another chance! Thank you for your patience and kindness, and wish you all the best!

Thank you for your consideration!

Sincerely,

Yating Peng

Hunan Agricultural University

[email protected]

December 14, 2022
